# Integrated Hydrolysis of Mixed Agro-Waste for a Second Generation Biorefinery Using *Nepenthes mirabilis* Pod Digestive Fluids

**Nkosikho Dlangamandla**, **Seteno Karabo Obed Ntwampe \***, **Justine Oma Angadam,** **Elie Fereche Itoba-Tombo, Boredi Silas Chidi** **and Lukhanyo Mekuto**

Bioresource Engineering Research Group (*BioERG*), Department of Biotechnology, Faculty of Applied Sciences, Cape Peninsula University of Technology, P.O. Box 652, Cape Town 8000, South Africa; dlangamandlans@gmail.com (N.D.); omajustine@gmail.com (J.O.A.); elie.tombo@gmail.com (E.F.I.-T.); boredi2002@gmail.com (B.S.C.); lukhayo.mekuto@gmail.com (L.M.)
* Correspondence: NtwampeS@cput.ac.za; Tel.: +27-214-609-097

**Abstract:** To sustainably operate a biorefinery with a low cost input in a commercial setting, the hydrolysis of lignocellulosic biomass must be undertaken in a manner which will impart environmental tolerance while reducing fermenter inhibitors from the delignification process. The challenge lies with the highly recalcitrant lignin structure, which limits the conversion of the holocelluloses to fermentable total reducing sugars (TRS). Due to these challenges, sustainable and innovative methods to pre-treat biomass must be developed for delignocellulolytic operations. Herein, *Nepenthes mirabilis* digestive fluids shown to have ligninolytic, cellulolytic and xylanolytic activities were used as an enzyme cocktail to hydrolyse mixed agro-waste constituted by *Citrus sinensis* (orange), *Malus domestica* (apple) peels, cobs from *Zea mays* (maize) and *Quercus robur* (oak) yard waste. The digestive fluids contained carboxylesterases (529.41 ± 30.50 U/L), β-glucosidases (251.94 ± 11.48 U/L) and xylanases (36.09 ± 18.04 U/L), constituting an enzymatic cocktail with significant potential for the reduction in total residual phenolic compounds (TRPCs), while being appropriate for holocellulose hydrolysis. Furthermore, the maximum TRS obtainable was 310 ± 5.19 mg/L within 168 h, while the TRPCs were reduced from 6.25 ± 0.18 to 4.26 ± 0.09 mg/L, which was lower than that observed when conventional methods were used. Overall, *N. mirabilis* digestive fluids demonstrated an ability to support biocatalytic processes with minimised cellulases hydrolysis interference. Therefore, the digestive enzymes in *N. mirabilis* pods can be used in an integrated system for feedstock hydrolysis in a second generation biorefinery.

**Keywords:** agro-waste; biorefinery; β-glucosidase; carboxylesterases; cellulases; *Nepenthes mirabilis*; total reducing sugars; xylanase

## 1. Introduction

The challenge associated with environmental pollution and petroleum depletion has ensured interest in the production of biofuels including value-added products using environmentally benign processes. The production of such commodities has demonstrated the feasibility and applicability of biorefinery processes that are microbially mediated, using lignocellulosic biomass feedstock, such as agro-waste, as a primary source of fermentable constituents. Moreover, the biorefinery processes consist of several stages, i.e., processing units, which include the preparation and hydrolysis of the feedstock, i.e., lignocellulosic biomass including agro-waste, in the upstream processes, after fermentation and the recovery of bio-products in other downstream processes. Recent studies have investigated

numerous processes to hydrolyse lignocellulosic biomass including agro-waste for biorefineries [1]. In these processes, the evaluation focused on reduced hydrolysis time for maximising the extraction of fermentable carbohydrates, reduced energy intensity, environmental benignity by eradicating inorganic compound usage and minimisation of operational costs [2]. These processes, albeit achieving varying successes, include chemical, physical, biological and physico–chemical hydrolysis technologies, either as individualised and/or amalgamated processes [3–5].

Despite the recent successes in lignocellulosic biomass hydrolysis using the processes above, several challenges still prevail. Currently, biomass hydrolysis is conducted in two to four stages that include enzyme hydrolysis, depending on the desired outcomes. However, many biomass hydrolysis stages can increase the operational costs of a biorefinery. Furthermore, hydrolysis, as it is used for the delignification of biomass to extract fermentable carbohydrates, culminates in the production of inhibitors, which may inhibit enzymatic hydrolysis and subsequently, downstream fermentation processes [6–9]. This necessitates further improvement of hydrolysis methods that are currently in use since the hydrolysis processes are one of a few costly processes in a biorefinery. These improvements can focus mainly on maximising the production of extractable fermentable sugars referred to as total reducing sugars (TRS) while reducing inhibitors and minimising operational costs, which could be vital to the success of a biorefinery even in developing countries.

Currently, an integrated (single stage/unit/reactor) and an environmentally benign process, has not been developed for the hydrolysis of feedstock for a second generation biorefinery, particularly focusing on the use of a renewable bioresource, such as mixed agro-waste. These processes are necessary for an effective and efficient way to resolve challenges associated with, (1) delignification, (2) cellulolysis and (3) the production of recalcitrant residual inhibiting by-products from the biomass hydrolysis process, all of which have an effect on other downstream processes [10].

To achieve this, process integration is required to improve and reduce some of the stages in the hydrolysis of biomass, i.e., using a suitable single pot multi-reaction process for biodelignification, cellulolysis and the reduction of inhibitory by-products from the feedstock. For such a strategy to succeed, digestive enzymes, such as those produced by *Nepenthes mirabilis*, which have been previously associated with delignification and cellulolysis [11] are required, with the potential to biodegrade delignification inhibitors.

The *Nepenthes* plants are carnivorous and use specialised pitfall traps like the pitcher plant, which traps insects and decomposes organisms that are perceived to be indigestible as they consist of fibrous chitin [12]. Most pitcher plant species produce an acidic fluid, with a pH ranging from 1.5 to 6 depending on the species [13]. Although numerous studies have been conducted to profile the constituents of such acidic fluids, there is minimal information on the pitcher plant's acidic fluid (extract) usability in novel processes [14], and their ability to facilitate the biodegradation of lignin, including the conversion of holocelluloses in biomass. However, recent studies have addressed some previously unknown information, indicating that a diverse and complex enzymatic cocktail does exist in *N. mirabilis* pods, with a high concentration of digestive/hydrolytic enzymes [11]. Chan et al. [11] reported that the biodegradation of the insects includes complex chitinolytic, proteolytic, amylolytic, cellulolytic and xylanolytic enzymes.

Generally, *N. mirabilis* digestive fluids, i.e., from the pitcher plants 'monkey cup', have been reported to contain digestive enzymes which are capable of biodegrading complex and polymeric molecules, such as glycan, starch and elemental metallic species, even under anaerobic conditions [11]. The application of such digestive enzymes in a biorefinery can minimise energy requirements, plant footprint and the use of hazardous chemical compounds while reducing production of fermentation inhibitors. These qualities make such digestive fluids suitable for a single pot (batch) multi-reaction biomass hydrolysis system, thus, an integrated process that can advance process capacity in the biorefinery industry, a daunting assertion when mixed agro-waste is to be considered as a feedstock. Overall, *N. mirabilis* (pitcher plant) digestive fluids have not been explored for the hydrolysis of lignocellulosic biomass destined for biorefineries, although the digestive fluids (1) have the ability to

decompose a variety of polymeric substances [12]; (2) are acidic [15], thus, can be used directly for the hydrolysis of biomass, a process analogous to diluting acid hydrolysis; and (3) contain microbial populations of the phyla Proteobacteria, Bacteroidetes, Actinobacteria, Verrucomicrobia and Planctomycetes, which are associated with the production of numerous hydrolytic enzymes [11].

Therefore, the primary aim of this study was to evaluate an integrated hydrolysis of mixed agro-waste destined for a second generation biorefinery using *N. mirabilis* digestive fluids. Such research is required to ascertain the applicability and biocatalytic efficacy of *N. mirabilis* digestive fluids in an integrated process for biomass hydrolysis of mixed agro-waste to maximise the extraction of TRS. The study also hypothesised that conventional methods, i.e., hot water, diluted acid and cellulases (sequential or combined) can be replaced by an *N. mirabilis* digestive fluid process in combination with commercial cellulases for the hydrolysis of mixed agro-waste.

## 2. Materials and Methods

### 2.1. Lignocellulosic Biomass Feedstock: Mixed Agro-Waste

#### 2.1.1. Collection and Preparation

Feedstock, i.e., agro-waste constituted with *Citrus sinensis* (orange), *Malus domestica* (apple) peels, cobs from *Zea mays* (maize) and *Quercus robur* (oak) yard waste, was collected from an agricultural produce market in the vicinity and the garden of the District 6 campus of the Cape Peninsula University of Technology (CPUT) (Western Cape, Cape Town, South Africa) respectively, after drying at 80 °C for 24 h and pulverisation (>45 μm to <100 μm) without a pre-rinsing step. The *C. sinensis* peels were re-dried further for 48 h and subsequently re-pulverised. These agro-wastes were used as mixed agro-waste feedstock, using a 1:1 ratio, i.e., 25% (*w/w*) for each; 1 g oranges, 1 g apples, 1 g maize cobs and 1 g oak, since there is minimal knowledge about the biodelignification of such mixed agro-waste, with most research studies focusing on the use of a single feedstock. The dried agro-waste (2 g) was slurried in sterile distilled water (sdH$_2$O, 200 mL) using airtight multiport Erlenmeyer flasks (250 mL, triplicate) fitted with a sampling syringe, corresponding to a 1% (*w/v*) slurry [3], at ambient temperature (25 to 30 °C) in a shaking (120 rpm) incubator (LABWIT ZWY-240, Shanghai Zhicheng Analytical, Shanghai, China), to ensure homogenisation. The pH was measured using a pH meter (Lasec Pty Ltd, Ndabeni, South Africa), at different stages of the experiments.

#### 2.1.2. Mixed Agro-Waste Lignin and Holocelluloses Content

The lignin and holocelluloses content of the mixed agro-waste was determined for un-hydrolysed, and *N mirabilis* hydrolysed samples with further hydrolysis using commercial cellulases to compare differences in the efficacy of techniques used for hydrolysing the biomass. The solids samples from the hydrolysis steps were dried at 80 °C in an oven for 24 h to reduce the moisture content [16]. After that, the samples were cooled to ambient temperature and stored under dry conditions. The analyses were carried-out by slurring 300 mg of dried hydrolysed agro-waste in 100 mL Schott bottles, with 1 mL of 72% of H$_2$SO$_4$ (Sigma-Aldrich, St. Louis, MO, USA), after agitation with a glass rod until the agro-waste was homogenously mixed [17]. After that, the Schott bottles were placed at 30 $\pm$ 0.5 °C using a water bath for 1 h. To further treat the samples, 28 mL of sterile distilled water (sdH$_2$O) was added after autoclavation at 121 °C for 30 min. The solution was cooled to 80 °C and filtered with the pre-weighed fibre glass filter [18]. The filtrates were transferred into 50 mL conical tubes and the acid-soluble lignin was determined at 205 nm using a Jenway 7305 UV/Vis spectrophotometer (Cole-Parmer, Staffordshire, UK), whereby the extinction coefficient of lignin (110 g/L·cm$^{-1}$, according to TAPPI UM 250) was used to quantify the acid-soluble lignin using Equation (1).

$$ASL = \frac{A \cdot D \cdot V}{a \cdot b \cdot M} \tag{1}$$

where *A* is the absorption at 205 nm, while *D* is the dilution factor, *V* is the volume of the filtrate (i.e., 0.029 L), *a* is the extinction coefficient of lignin (i.e., 110 g/L·cm$^{-1}$, according to TAPPI UM 250), *b* is the cuvette path length, in cm (i.e., 1 cm) and *M* is the oven dry weight of the sample (i.e., 100% dry matter) before acid digestion.

Furthermore, the residues were washed with hot water, and a mixture of ethanol (Sigma-Aldrich, St. Louis, MO, USA), benzene (Sigma-Aldrich, St. Louis, MO, USA) and ether until the residues were contaminant free [16]. The residue was dried at 105 °C in the oven for 2 h, to quantify the total lignin content as the difference between the sum of both acid soluble and insoluble lignin. The residues on the fibre glass filter papers were transferred to crucibles and calcined at 700 °C for 1 h in a furnace (Labofurn furnace, model EMF260, Kiln Contracts (Pty) Ltd., Cape Town, South Africa) to quantify total ash determined by the mass difference of the ash content of acid-insoluble lignin and ash content of the filters without the residues. This method was applied to all samples in triplicate. The total ash content was determined accordingly using Equation (2).

$$sh(\%) = \frac{M_c}{M_a} \times 100 \tag{2}$$

where *Ash (%)* is the percentage by mass of ash, $M_c$ is the ash mass (mass difference between crucible filter with residue, crucible with control filter and empty crucible) and $M_a$ is the mass of the dry sample.

### 2.1.3. Inhibitory Compound Quantification: Total Residual Phenolic Compounds (TRPCs)

The analyses were conducted by transferring 5 mL of the slurried mixed agro-waste into 15 mL conical tubes after centrifugation at $4000\times g$ for 5 min with a volume (3 mL) of the recovered supernatant being diluted with an equivalent volume (3 mL) of sdH$_2$O in clean 15 mL conical tubes, while the remaining volume (1 mL, undiluted) was used for TRS analyses (Section 2.3). After that, the following analyses were periodically conducted during the hydrolysis process, taking into account the dilution factor.

Phenolic compounds in agro-waste are known as inhibitors of β-glucosidase [19], which are primarily responsible for facilitating the hydrolysis of oligosaccharides to fermentable sugars [20]. Therefore, TRPCs were quantified using the Folin–Ciocalteu method [21], with a volume (100 μL) of the diluted supernatants being added to an assay mixture containing sdH$_2$O (1.5 mL) and the Folin–Ciocalteu reagent (250 μL, Merck KGaA, Darmstadt, Germany) after the addition of sodium carbonate (1 mL, 20% *w/v*, Merck Chemicals (PTY) LTD, Sandton, South Africa) after 3 min. After that, the assay mixture was homogenised in darkness for 1 h. A volume (1 mL) of the assay mixture was then analysed using plastic cuvettes (1.5 mL) at 650 nm using a Jenway 7305 UV/Vis spectrophotometer (Cole–Parmer, Staffordshire, UK), with the TRPCs' concentration being determined using 2 to 10 mg/L of 1,2-dihydroxybenzene (Sigma-Aldrich, St. Louis, MO, USA) in sdH$_2$O for the calibration curve [22,23].

### 2.2. Nepenthes Mirabilis Digestive Fluids

#### 2.2.1. Collection and Preparation

Pooled digestive fluids from *N. mirabilis* plants grown in a greenhouse under controlled conditions were collected from Pan's Carnivores Plant Nursery (21 Kirstenhof, Tokai, Cape Town, South Africa). The digestive fluids were collected using sterile 50 mL conical tubes and immediately stored on ice, prior to transportation to the laboratory, where they were centrifuged at $4000\times g$ for 15 min and filter sterilised with 0.22 μm Millipore membranes (Merck, Burlington, MA, USA) with subsequent storage at 4 °C prior to use, i.e., without dilution or the use of a buffer. About 15 to 35 mL of the digestive fluids were collected per 'monkey cup' of the *N. mirabilis* plants, depending on the size of the cup.

2.2.2. *N. mirabilis* Digestive Fluids Characterisation

Physico–Chemical Characteristics of the *N. mirabilis* Digestive Fluids

The physico–chemical characteristics of the *N. mirabilis* digestive fluids, such as pH, conductivity, specific gravity and redox potential, were determined using a multi-parameter meter (Eutech Instruments PTY Ltd/Thermo Fisher Scientific, Queenstown, Singapore) as highlighted in Table 1. The redox potential and conductivity of the digestive fluids were measured to verify the ability of the *N. mirabilis* digestive fluids to facilitate oxidation reactions and electrolytic strength [24], respectively. The specific gravity and pH were measured to evaluate the density and acidity of the digestive fluids [4], respectively.

**Table 1.** Physico–chemical characteristics of the *Nepenthes mirabilis* digestive fluids.

| Characteristic/Parameter | Values (units) |
|---|---|
| pH | 1.80–2.2 |
| Specific gravity ($\varrho_{extracts}/\varrho_{H_2O}$) | 0.67–0.82 |
| Redox potential (Eh) | 510–526 mV |
| Conductivity | 3.5–5.89 mS/cm |

Microbial Identification and Biocatalytic Activity of the *N. mirabilis* Digestive Fluids

Before filtration, microbial population identification was initially done using a VITEK 2 system V07:01 (BioMérieux, Craponne, France) utilising Gram-negative cards (GN cards) and Gram-positive cards (GP card) as per the manufacturer instructions [25]. Further identification of microbial strains in the digestive fluids was performed using a DNA extraction method including sequencing. For DNA extraction, the cultures were cultivated by adding 1 µL of the *N. mirabilis* digestive fluids into 15 mL glass test tubes that contained Luria broth (5 mL) after incubation (37 °C) for 24 h. After that, the cultures were inoculated on Luria Bertani agar (LBA) plates at 30 °C for 24 h, with single colonies being sub-cultured for species purification and identification [11]. A staining procedure was also performed for each isolate for morphological assessments.

In this study, DNA extraction and Polymerase Chain Reaction (PCR) amplification of 16S ribosomal deoxyribonucleic acid (rDNA) was performed in an external laboratory (i.e., Inqaba biotech, Pretoria, South Africa) using the commercial genomic DNA purification kit (Zymo Research; Fungal/Bacterial DNA kit, Irvine, CA, USA), as per the manufacturer's instructions. The genomic DNA of strain was extracted for PCR using universal bacterial primers targeting the 16S rDNA gene [26]. The DNA was assessed using (1) a 0.5% (*v*/*v*), i.e., 500 µL per 100 mL, of the genomic lysis buffer, while the cell disruptor was processed at maximum speed for 5 min, subsequently centrifuged at $10,000 \times g$ for 1 min in a lysis tube; (2) A volume (400 µL) of supernatant of Zymo-Spin was transferred into a spin filter in a collection tube and was also centrifuged at 7000 rpm for 1 min; (3) A volume (1.2 mL) of Fungal/Bacterial DNA binding buffer was added to the filtrate in the collection tube from Step 2; (4) A volume (800 µL) of the mixture from Step 4 was transferred to a Zymo-Spin column in a collection tube and centrifuged at $10,000 \times g$ for 1 min. The filtrate was discarded, and the process was repeated; (5) A volume (200 µL) of DNA pre-wash buffer was added to the Zymo-Spin column and centrifuged at $10,000 \times g$ for 1 min, and then washed with 500 µL of Fungal/Bacterial DNA Wash Buffer.

The mixture in the column was transferred to a clean 1.5 mL micro-centrifuge tube, and a volume of 100 µL DNA elution buffer was directly added to the column matrix, after being centrifuged at $10,000 \times g$ for 30 s to elute the DNA. The PCR amplification was conducted using primers 27F (5′-AGAGTTTGATCMTGGCTCAG-3′) and 1492R (5′-GGTTACCTTGTTACGACTT-3′), which are universal primes for 16S rDNA. PCR was conducted in 100 µL reactions, while 100 ng of genomic DNA was used [27]. The PCR conditions were set at 36 cycles of 98 °C denaturation for 30 s, primer annealing at 60 °C for 20 s, and elongation at 72 °C for 60 s. PCR products were further gel extracted (Zymo Research, Zymo Clean Tm Gel DNA Recovery kit, Irvine, CA, USA), while typing was done by capillary

electrophoresis in an ABI PRISM 3500xl Genetic analyser (Eutech Instruments PTY Ltd/Thermo Fisher Scientific, Queenstown, Singapore). The PCR products were further purified using Zymo Research, ZR-96 DNA Sequencing Clean-up kit (Irvine, CA, USA) and analysed using a CLC main workbench (QIAGEN, Redwood City, CA, USA). After that, 16S rDNA was compared with available nucleotide sequences in NCBI Genbank database (http://blast.ncbi.nlm.nih.gov/Blast.cgi), for the identification of strains that are available in the *N. mirabilis* pod digestive fluids with the following accession number *KY249126.1*, *DQ513324.1* and *KU948294.1* [28].

Specific enzymes were selected to quantify their activity in *N. mirabilis* digestive fluids based on their efficacy to hydrolyse the agro-waste constituents, including the by-products formed. Enzymes of interest, namely β-glucosidases, xylanases and carboxylesterases, were previously identified as (1) being an essential cellulose biodecomposing component facilitating the penultimate bottleneck for biocatalytic conversion of cellobiose, a reducing sugar, to glucose [11], (2) having the potential to biodegrade and, thus, solubilise hemicellulose [29] and (3) having the potential as a candidate phenolic acid esterases with a hydrolytic activity against carboxylester bonds between holocelluloses sugars and lignin [30].

As such, β-glucosidase activity in the *N. mirabilis* digestive fluids was quantified using ϱ-nitrophenyl-β-D-glucopyranoside (pNPG) as a substrate [31], with the rate of formation for xylose, a reducing sugar from xylan, also being determined from xylanase assays using xylan [32–34]. Similarly, carboxylesterase activity was determined using ϱ-nitrophenyl acetate (pNPA) as the substrate [30,35,36]. All these assays were performed using a *N. mirabilis* digestive fluid in an appropriately buffered mixture at ambient temperature, using a Cecil 2021 UV/Vis spectrophotometer (Cecil Instruments, Cambridge, UK) set in kinetics mode (refer to Table 2) for experimental conditions. Overall, the activity quantified was based on the concentration of product formed per min (U/L), with compounds being calculated by Equation (3) and Table 2, which highlights the enzymatic assay conditions. A summarised schematic flow diagram of the microbial identification and biocatalytic process is shown in Figure 1.

$$\text{Enzyme activity (U/L)} = \left[ \left( \frac{\frac{dA}{dt} \cdot D_f}{\varepsilon} \right) \times 60 \times 10^6 \right] \tag{3}$$

where *dA/dt* is the initial rate reaction, while the $D_f$ is the dilution factor and $\varepsilon$ is the extinction coefficient.

**Table 2.** Enzyme activity assays, reagents and conditions for *N. mirabilis* digestive fluids

| Reagent/Parameter | 1,4-β-glucosidase | Endo-Xylanase | Carboxylesterase |
|---|---|---|---|
| Substrate concentration | 0.35 mM pNPG [#] | 54.2 mM Xylan | 0.5 mM pNPA [#] |
| Substrate volume | 0.8 mL | 1.8 mL | 0.8 mL |
| Product formed | *p*NP | Xylose | *p*NP |
| Buffer(s) & volumes | 50 mM sodium acetate, pH 6, 600 μL | 100 mM McIlvaine, pH 5, 1600 μL | 100 mM Tris-HCL, pH 7.8, 200 μL |
| Volume of enzyme | 200 μL | 200 μL | 300 μL |
| Temperature | 25 °C | 25 °C | 25 °C |
| Wavelength (kinetics mode) | 410 nm | 586 nm | 410 nm |
| Extinction coefficient ($\varepsilon$) | 18,100 M$^{-1}$·cm$^{-1}$ | 135 M$^{-1}$·cm$^{-1}$ | 17,000 M$^{-1}$·cm$^{-1}$ |

[#] final concentration in the assay mixture, Tris-HCL—tromethamine hydrochloride buffer, McIlvaine—citrate-phosphate buffer.

| Microbial identification |
|---|

| |
|---|
| 1. Gram staining |
| 2. DNA extraction |
| 3. Amplification: Polymerase chain reaction (PCR) |
| 4. DNA purification |
| 5. DNA sequencing |

| Biocatalytic activity of the *N. mirabilis* digestive fluids |
|---|

| |
|---|
| Enzyme assay |
| 1. β-glucosidase |
| 2. Xylanase |
| 3. Carboxylesterase |
| |
| Quantification: Kinetics mode: Cecil 2021 UV/Vis spectrophotometer |

**Figure 1.** A schematic flow diagram of the microbial population identification and biocatalytic activity of the *Nepenthes mirabilis* digestive fluids.

2.2.3. Mixed Agro-Waste Hydrolysis Procedure Using *N. mirabilis* Digestive Fluids and Sequential Commercial Cellulases Hydrolysis

The hydrolysis of the mixed agro-waste followed a sequence whereby the mixed agro-waste was slurried (see Section 2.1.1) for 72 h to solubilise some of the constituents in the waste, prior to the direct supplementation of the pooled *N. mirabilis* digestive fluids into each homogenised Erlenmeyer flasks after further (96 h) ambient temperature incubation for a total experimental time of 168 h. For the furthering of cellulolysis for the slurried agro-waste, cellulases (SAE 0020-Sigma-Aldrich, Darmstadt, Germany) supplementation, a commercial product that contains cellulases β-glucosidases and hemicellulases, was performed after 48 h post *N. mirabilis* supplementation for each Erlenmeyer flask, i.e., sequential to the hydrolysis using *N. mirabilis* digestive fluids. The mixed agro-waste was further treated with cellulases after the slurrification procedure and supplementation with *N mirabilis* digestive fluids, i.e., after an additional 48 h, at a stage where the experiment was at 120 h from its initiation. A volume (1200 μL) of cellulases (24.67 U/mL) was added to each flask, constituting 600 μL cellulases/g agro-waste [37], without pH correction, with requisite analyses being conducted. This process was conducted in a single pot (batch), multiple reaction system. After each intermediate hydrolysis stage, agro-waste-free samples (5 mL) were collected for various analyses (TRPCs) by initially centrifuging ($4000\times g$ for 5 min), the supernatant containing agro-waste biomass using a preparatory strategy analogous to that reported in Section 2.1.3. After that, the agro-waste pellets were thoroughly rinsed with sdH$_2$O and air-dried at room temperature to reduce the moisture content before the assessment of structural modifications using Fourier Transform Infra-red Spectroscopy (FTIR) and powder X-ray diffraction (XRD) systems for each of the agro-waste samples recovered. Control experiments using untreated mixed agro-waste served as a reference [38].

### 2.2.4. Hydrolysis of the Mixed Agro-Waste Using Hot Water, Dilute Acid and Cellulases

The hot water, dilute sulphuric acid (Sigma-Aldrich, St. Louis, MO, USA), and cellulases hydrolysis were conducted in this study for comparison as combined conventional methods. Hot water hydrolysis was done using biomass (2 g, mixed agro-waste) at a high temperature (120 °C) in an autoclave for 15 min in 250 mL Schott bottles; whereby 200 mL of distilled (sdH$_2$O) water was added. Recent studies have concluded that the hot water treatment is effective at temperatures between 120 and 200 °C [39]. After that, the mixture of the agro-waste was cooled to ambient temperature. Aliquots (5 mL) were sampled into sterile 15 mL conical tubes followed by centrifugation at 4000× *g* for 5 min to recover sedimented agro-waste biomass. A volume (1 mL) of aliquots was diluted with 9 mL of sdH$_2$O for TRS analysis. A volume (4 mL) of the aliquots withdrawn was subsequently returned to the Schott bottles. The recovered pellets of mixed agro-waste after centrifugation were dried at 80 °C for 24 h and kept at ambient temperature for further analysis. The dilute acid hydrolysis ensued thereafter, with a concentration of diluted sulphuric acid, i.e., 1% (*v/v*), as a final mixture concentration, being added directly to the hot water hydrolysis Schott bottles to ascertain a single pot batch system [40], after autoclavation at 121 °C for 15 min [39]. The mixture was analysed for TRS before proceeding to cellulases hydrolysis, whereby further hydrolysis of the mixed agro-waste was done by cellulases (24.67 U/L) supplementation. A volume of cellulases (600 µL) per g of mixed agro-waste was added to the reaction mixture to further enhance hydrolysis outcomes at 55 °C and pH 4.5 for 72 h; a pH which was attained using sodium acetate buffer.

### 2.3. Determination of Total Reducing Sugars (TRS)

The TRS produced during hydrolysis of the agro-waste were quantified by using a 3.5 dinitrosacylic acid (DNS) reagent (Sigma-Aldrich, St. Louis, MO, USA), composed of DNS (10 g), phenol (2 g, Sigma-Aldrich, St. Louis, MO, USA), sodium sulphite (0.5 g, Sigma-Aldrich, St. Louis, MO, USA) and sodium hydroxide (10 g, Sigma-Aldrich, St. Louis, MO, USA) made up to 1 L; whereby the prepared samples (1000 µL), as highlighted in Section 2.1.3, were diluted in 9 mL of sdH$_2$O. The assay mixture contained 1.5 mL of the diluted aliquots, 1500 µL DNS reagent in sterile 15 mL test tubes after heating up to 90 °C for 10 min. The assay mixture was cooled to ambient temperature before the addition of 0.5 mL of 40% (*w/v*) sodium potassium tartrate solution. The absorbance was determined using a Jenway 7305 UV/Vis spectrophotometer (Cole–Parmer, Staffordshire, UK) at 575 nm. The reference was analysed using a similar procedure without the TRS containing samples, i.e., replacing the sample volume with sdH$_2$O, with different glucose concentrations being used to generate a suitable calibration curve [41].

### 2.4. Powder X-ray Diffraction Analyses

To determined crystallinity of the agro-waste, an un-hydrolysed and post-hydrolysis mixed agro-waste were analysed using an XRD (Bruker Pty Ltd, Sandton, South Africa) at 40 kV and 40 mA with a D2 phaser with a Lynxeye, providing a suitable peak-to-background ratio [42]. The scanning range (2$\theta$) was 10 to 50° at a ramping scale of 0.017°, using zero background holder plates (50 µm depth), with the crystallinity index (*CrI*) being determined using Equation (4) with residual mixed agro-waste (5 mg) being used.

$$CrI(\%) = \frac{(\Delta I)}{I_{002}} \times 100 \tag{4}$$

where by $\Delta I$ is ($I_{002} - I_{am}$), while $I_{002}$ is the intensity for the crystalline portion of the mixed agro-waste (i.e., un-or hydrolysed) at 2$\theta$ between 21 and 22°, where $I_{am}$ is the peak for the amorphous portion of the mixed agro-waste (i.e., un or hydrolysed) at a range of 2$\theta$ between 14 and 19° [43].

## 2.5. Fourier-Transform Infrared Spectroscopy Analysis

Functional group modifications in the mixed agro-waste were determined to assess the effectiveness of the hydrolysis methods being studied, using an $\alpha$-FTIR spectrometer (Bruker Pty Ltd, Sandton, South Africa) and a Smart iTR with a diamond crystal window. A mass of the un- and hydrolysed mixed agro-waste samples mixed with KBr (Sigma-Aldrich, St. Louis, MO, USA) was placed in the diamond crystal window of the Smart iTR. Initially, the measurements were taken against a background spectrum of the diamond window without the mixed agro-waste. The spectra scans were collected from a range of 400 to 4000 cm$^{-1}$ with a spectral resolution of 4 cm$^{-1}$ at 100 scans per min [44].

## 2.6. Experimental Data Handling, Computations and Statistical Analysis

To implement the single pot (batch) multi-reaction hydrolysis process of the mixed agro-waste, a total experimental run time of 168 h was implemented, with periodic sampling for various analyses at 24, 48, 72, 96, 120 and 168 h, prior to the supplementation of *N. mirabilis* digestive fluids (72 h) and cellulases (120 h). The data generated were analysed to determine the mean value and standard error of the mean (SEM) for the raw data obtained. All experimental data were computed to take into account sample dilutions to quantify the actual concentrations for parameters monitored. Furthermore, all experimental analyses were conducted in triplicate, i.e., n = 3 samples per experiment and analyses with the SEM being determined using Equation (5).

$$EM = \frac{\text{Standard deviation}}{\sqrt{\text{Number of sample tested}}} \tag{5}$$

For FTIR and XRD, the recovered air-dried agro-waste was pooled to attain a composite sample, which was then used for analysis.

## 3. Results and Discussion

### 3.1. Selection of Agro-Waste

The mixed agro-waste feedstock selected was based on its regional availability in South Africa, especially in the Western Cape, South Africa. Overall, citrus and oak trees are largely available, focusing on the specification requirements of the feedstock to be used in a second generation biorefinery. Second generation biorefinery feedstock may include wood waste, non-food crops, waste cooking oil and forestry agricultural residues [45]. These types of feedstock reduce the reliance on edible crops [28]. The Western Cape Province (SA) is the third largest province that produces large quantities of citrus fruit in South Africa [46], including 95% of apples (*M. domestica*). Furthermore, oak trees (*Q. robur*) generate a large quantity of yard waste with the tree being able to live for 300–600 years. Therefore, the agro-waste was selected to reduce landfilling, by converting it into fermentable sugars.

In previous studies, agro-waste, in particular, *C. sinensis*, *M. domestica*, corn cobs from *Z. mays* and *Q. robur*, had been determined to contain sufficient quantities of extractable fermentable sugars [26,47–49]. Although some produce inhibiting by-products, such as ferulic, glucuronic, p-coumaric, acetic acids and phenolics including residual heavy metals during hydrolysis [10], some of which were hypothetically assumed to be biodegradable using *N. mirabilis* digestive fluids. To further extract fermentable sugars from hydrolysed agro-waste, further potential hydrolysis of the residual hydrolysed agro-waste using commercial cellulases must be feasible, even when a different cocktail of lignocellulolytic enzyme is used [20], in particular, an enzyme cocktail which has properties, such as those observed in *N. mirabilis* plant's digestive fluids [11].

From Table 3, the increment in lignin content in residual hydrolysed agro-waste, in comparison to the reduction in holocellulose, was directly attributed to the hydrolysis regime implemented; albeit that it was observed that the recalcitrant acid-insoluble lignin was ineffectively decomposed

by the enzymatic hydrolysis regime used. This is an advantageous attribute as the system seemed to only decrease holocellulosic content of the biomass feedstock used and suggests a reduction in the production of phenolics from the agro-waste.

**Table 3.** Residual lignin and holocellulose content of the untreated *N. mirabilis* plant digestive fluids/cellulases and hot water/dilute acid/cellulases hydrolysed agro-waste composite samples.

| Hydrolysis Methods | Residual Lignin (%) | Residual Holocellulose (%) | Ash (%) |
|---|---|---|---|
| Untreated mixed agro-waste | 27 | 72.9 | 0.1 |
| *N. mirabilis* [#] | 39 | 60.7 | 0.1 |
| *N. mirabilis*/Cellulases | 59 | 40.7 | 0.3 |
| Hot water/Dilute acid/Cellulases | 43 | 56.8 | 0.2 |

[#] *Nepenthes* pitcher plant's digestive fluids.

Total residual lignin constituted by acid-insoluble residue (AIR) and acid-soluble lignin (ASL) were determined by the Klason lignin method for untreated and *N. mirabilis* plant digestive fluids hydrolysed mixed agro-waste, which showed significant increases of residual lignin content in cellulases supplemented systems. Overall, lower lignin concentration and its composition contribution were observed to be 27 to 39% when the agro-waste was hydrolysed solely with *N. mirabilis* plant digestive fluids, while the holocelluloses were 72.9% and 60.7%, respectively. These results illustrated a potential extraction of some holocelluloses during hydrolysis, which was further reduced to 40.7% (Table 3), when commercial cellulases were supplemented in *N. mirabilis* hydrolysis, culminating with the residual mass being highly crystalline with a large proportion of unreactive ash.

Therefore, the hydrolysed samples showed recoverable and/or convertible holocelluloses content using the *N. mirabilis*/cellulases hydrolysis regime. Generally, with the *N. mirabilis* plant digestive fluids hydrolysis, a larger fraction of the lignin content in agro-waste remained as acid insoluble residue. On the other hand, for conventional acid-based hydrolysis, acid soluble by-products were observed, which can lead to inhibition of subsequent and downstream processes. By using the *N. mirabilis* plant digestive fluids, the hydrolysis of the agro-waste was hypothesised to enhance fermenter performance with minimal inhibition. By sequentially using conventional methods, i.e., hot water, dilute acid and cellulases hydrolysis in a single reactor system, a large proportion of lignin seemed to have been biodegraded into solution, leaving a higher quantity of holocelluloses (56.8%) intact in the residual biomass compared to the *N. mirabilis'* subsequent cellulases hydrolysis system.

*3.2. Biophysico–Chemical Characteristics of the N. mirabilis Digestive Fluids*

The dominant bacteria in *Nepenthes* pitcher digestive fluids were *Klebsiella oxytoca* (KF55591), *Bacillus thuringiensis* (KF557957); although, *Bacillus cereus* and *Bacillus anthracis* were not identified with the Vitek method as shown in Table 4. The most abundant enzymes in the pitcher plant's digestive fluids were identified as proteases, nucleases, peroxidases, chitinases, phosphatases and glucanases, including carboxypeptidases [12]. Lee et al. [12] also identified these proteins, whereby the carboxypeptidases and prolyl endopeptidases were identified as novel proteins. The enzymes that were observed in the *Nepenthes* sp. digestive fluids have been shown to be stable under acidic (i.e., low pH) oxidative and low nitrogen conditions [12]. Similarly, it is expected that bacteria producing these enzymes are able to thrive under such conditions while sustaining an environment that facilitates the degradation of complex carbohydrates, such as starch, xylan, hemicellulose and cellulose [11]. In previous studies, *Klebsiella oxytoca* was determined to be chitinolytic, with *Bacillus thuringiensis* (H1M), *Bacillus* H1a (*Bacillus* sp.) being shown to have proteolytic, amylolytic and cellulolytic activity; producing glycoside hydrolases, which are constituted by chitanase and glucanases [50]. For this study, there was evidence of microbial proliferation in the *N. mirabilis* pods' digestive fluids, with the dominant microbial species being *Bacillus* sp., i.e., *B. cereus*, *B. thuringiensis* and *B. anthracis,* including

*Klebsiella oxytoca,* which was also identified. The evaluation of the gastro-intestinal bacterial population in termites, determined for the decomposition of holocelluloses included *B. cereus, B. thuringiensis* and *B. subtilis,* which have been determined to be cellulases and xylanases producers with varying biocatalylic activity [51]. However, the functionality of these microorganisms at pH 2 and Eh + 500 mV is unknown.

**Table 4.** Identity of bacteria isolated from *N. mirabilis* digestive fluids.

| Identity | Vitek 2 | 16S rDNA | Accession Number |
|---|---|---|---|
| *Bacillus* sp. | + | − | u/a |
| *Klebsiella oxytoca* | + | − | u/a |
| *Bacillus cereus* | − | + | KY249126.1 |
| *Bacillus thuringiensis* | − | + | DQ513324.1 |
| *Bacillus anthracis* | − | + | KU948294.1 |

u/a—unassigned.

The redox potential (Eh) of the digestive fluids was measured to verify the ability of the *N. mirabilis* digestive fluids to accept or lose electrons, i.e., facilitate oxidation reactions. For highly effective oxidative reactions, a positive Eh of up to +810 mV under ambient and/or mild conditions (pH 7.0, 30 °C) is required [24]. In microbial systems, the Eh might be directly related to the growth of microorganisms, with aerobes being able to grow and proliferate in Eh from +300 to +500 mV, while anaerobes were determined to proliferate from −250 to +100 mV, with a slower growth rate occurring at higher Eh due to the highly oxidative environment which can culminate in oxidative species generation, which is known to be harmful to microbial cellular membranes [24]. Overall, the *N. mirabilis* pod's digestive fluids had an Eh averaging +510 mV, which was indicative of the oxidative properties of the digestive fluids [52].

Carboxylesterase, β-glucosidase, Xylanase and Commercial Cellulases Activity

Carboxylesterases are well known as acetylxylan esterases, which facilitate the hydrolysis of xylan. They facilitate the removal of ferulic acids from xylan including the bioconversion of holocelluloses [53]. In general, these esterases catalyse the hydrolysis of numerous acetyl groups in polymeric substances, such as xylan, acetylated xylose and acetylated glucose. The carboxylesterase activity was assessed as a hydrolysis-based reaction for the formation of 1 μmol of *p*-nitrophenol per min, which corresponded to an activity of 529.41 ± 30.57 U/L. Recent studies have shown carboxylesterase activity in *Penicillium chrysogenum*, whereby the activity was found to be 5.4 U/L [54], with *Aspergillus* and *Trichoderma* sp. being other organisms that were reported for carboxylesterase activity [55]. This illustrated the capability of the *N. mirabilis* pods' digestive fluids to hydrolyse acetyl groups including those associated with xylan. Furthermore, *Bacillus* sp., in particular, *B. vallismortis*, was confirmed to have the ability to produce short-chained acetyl xylan esterases [56]. The decoupling of the xylan backbone is highly dependent on the endo/β-xylosidases; albeit acetyl xylan esterases, including xylanases, are largely involved [55]. By reducing the acetyl groups, the endo-xylanases will, thus, be effective during the hydrolysis of mixed agro-waste [30], furthering the decomposition of xylan to xylose.

Furthermore, β-glucosidase activity (251.94 ± 11.48 U/L) was measured using the formation of pNPG per minute, resulting in similar results to those observed in the study by Kim et al. [50]. In this study, pNPG has been used as a substrate to quantify the activity of β-glucosidase using the pitcher plant's digestive fluids as active enzyme biocatalisers instead of cellobiases. Hydrolytic enzyme activities, including β-glucosidase, were positively quantified in numerous *Nepenthes* spp., with the activity of β-glucosidase disassociated with the influence of pH. Pitcher plants' fluid in *Nepenthes* sp. is determined to contain a higher activity of digestive enzymes, with a high concentration of natural antibacterial agents, and the bacteria from these fluids were considered to be robustly adaptive to even produce β-glucosidase, including xylanase. In addition, xylanase also has the ability to degrade

hemicellulose in agro-waste, in particular, β-1,4-xylan, with the *B. cereus* being determined to facilitate the decoupling of beta-bonds in xylan due to a xylanase mechanism [57].

### 3.3. Integrated Hydrolysis

#### 3.3.1. Total Residual Phenolics Content (TRPCs)

The total residual phenolic compounds were determined using the Folin–Ciocalteau reagent, as phenolic compounds inhibit the efficacy of β-glucosidase, thus, depletes the enzymes' functionality due their reactive species-scavenging properties, which can reduce the efficacy of these enzymes in an oxidative environment [22]. The highest TRPCs (6.25 ± 0.1 mg/L) were observed in slurrified mixed agro-waste supplemented with *N. mirabilis* digestive fluids (Table 5), with a decrease in TRPCs being observed after the addition of cellulases. The TRPCs concentration in untreated mixed agro-waste remained at 3.95 ± 0.12 mg/L throughout the 168-h experimentation period, an attribute imparted by insignificant reactions that were taking place in the untreated samples. A concentration of 5.65 ± 0.44 mg/L for TRPCs was also observed at the end of experiments conducted using combined conventional methods in a single pot (batch) multi-reaction hydrolysis system. The results obtained indicated that the presence of the *N. mirabilis* digestive fluids in the type of hydrolysis system designed, has an ability to reduce phenolic content generated, even if it is to a minimised extent. A similar study was conducted and reported [58] whereby the TRPCs were observed to vary from 26 to 34% of the total lignin loads in the hydrolysis systems used. This further confirmed that the mixed agro-waste has a large phenolic content, which can negatively affect downstream processes. Therefore, these results should be considered for the optimisation of feedstock preparation to minimise high phenolic content production.

**Table 5.** Total residual phenolic compounds in untreated, the *N. mirabilis* plant's digestive fluids/cellulases and hot water/dilute acid/cellulases hydrolysed agro-waste samples (SEM: standard error of the mean).

| Hydrolysis Methods | Sampling Time (h) | Total Phenolic Content (mg/L) [#] | SEM |
|---|---|---|---|
| Untreated mixed agro-waste | 168 | 3.95 ± 0.12 | 0.07 |
| *N. mirabilis* | 72 | 6.25 ± 0.18 | 0.11 |
| *N. mirabilis*/Cellulases | 168 | 4.26 ± 0.09 | 0.05 |
| Hot water/Dilute acid/Cellulases | 168 | 5.65 ± 0.44 | 0.25 |

[#] Different values represent the mean ± SD (n = 3).

#### 3.3.2. Performance of the Single Pot Multi-Reaction Hydrolysis Process

The performance of the single pot process designed for ambient temperature operations, using *N. mirabilis* digestive fluids, included the initial slurrification of the biomass, after *N. mirabilis* supplementation at 72 h, with the further addition of holocellulolysis, using commercial cellulases implemented at 120 h (Table 6). The effect of the pitcher plant's digestive fluids on the biodegradation of the mixed agro-waste was distinctively observed after 48 h of the supplementation of the *N. mirabilis* plant's digestive fluids. The results indicated that the concentration of the pitcher plant's digestive fluids had a significant effect on the degradation of holocelluloses with the maximum (max) TRS concentration measured being 310 ± 5.19 mg/L. The average TRS concentration (Table 6) illustrated the specific impact of the addition of *N. mirabilis* digestive fluids and further cellulases supplementation in a single pot system, with the maximum TRS achieved after 168 h. The effect of the cellulases on the hydrolysis process step was significant as illustrated by the increase of the TRS after 120 h. For conventional methods, an adequate increase from 1.4 ± 0.578 to 3.22 ± 0.219 g/L of TRS was observed in 72 h, which illustrated the impact of the cellulases hydrolysis. Furthermore, comparing the conventional methods and the proposed *N. mirabilis* supplementation of digestive fluids, similar

results were obtained for both methods. However, the *N. mirabilis* supplementation has further shown a reduction of inhibitor concentration during the process, while the conventional methods produced a higher concentration of inhibitors, i.e., TRPCs (Table 6).

**Table 6.** Average total reducing sugars (TRS) concentration of *N mirabilis* digestive fluids hydrolysis of mixed agro-waste at 72, 120 and 168 h.

| Hydrolysis Methods | Sampling Time (h) | Average TRS Concentration (mg/L) [#] | SEM |
|---|---|---|---|
| Untreated mixed agro-waste | 168 | 60.69 ± 1.7 | 1.02 |
| *N. mirabilis* | 72 | 244.91 ± 20.55 | 11.86 |
| *N. mirabilis* | 120 | 269.164 ± 18.94 | 10.94 |
| *N. mirabilis*/Cellulases | 168 | 310.55 ± 5.19 | 3.00 |

[#] Different values represents the mean ± SD (n = 3).

### 3.3.3. Furthering of Agro-Waste Hydrolysis Using Commercial Cellulases

To further evaluate the effect of the hydrolysis and to quantify the biodegradation suitability of the agro-waste using *N. mirabilis* digestive fluids, cellulases hydrolysis was performed. The process was completed after 72 h after the addition of cellulases, with a sharp TRS increase after 24 h when cellulases were added to the hydrolysis mixture. A maximum concentration of 310 mg/L was obtained (Table 6). The secondary function of *N. mirabilis* digestive fluids was to reduce the TRPCs while increasing TRS extraction in a single pot, batch system. This further biodegradation of the mixed agro-waste and reduction of inhibitors can be provided for by a suitable cocktail of enzymes, such as those in *N. mirabilis'* digestive fluids. Generally, the acid-based hydrolysis method has been associated with high-energy consumption and the production of inhibitors [9] and chemical usage, processes that are not environmentally benign [40].

### 3.3.4. X-ray Diffraction Analysis

The XRD analyses were conducted to analyse the deformation of the crystalline structure of the mixed agro-waste. Any biomass is constituted by cellulose, hemicellulose (i.e., holocellulose) and both are embedded in lignin [40]. Theoretically, the aim of hydrolysing agro-waste was to decouple structures that are bound together in the agro-waste for the ease of hydrolysis. Therefore, the *CrI* values were determined to vary depending on the effect of the hydrolysis processes used and in their ability to disintegrate the structure of the agro-waste, thus, reducing its crystallinity [59]. However, the crystalline determination of an entire lignocellulosic biomass can be difficult, since pooled samples were used as a representative of the crystalline structure of the biomass. In order for true crystallinity of holocellulosic material (mixed agro-waste) to be evaluated a comprehensive assessment is required. X-ray diffraction methods measuring the total crystallinity of lignocellulosic biomass, which includes the combined holocellulose and lignin, is used. Therefore, in this study, the *CrI* values of agro-waste were measured by the relative crystallinity of the holocelluloses for untreated and hydrolysed agro-waste [60]. As such, the *CrI* was determined by using the ratio between the intensity of the crystalline peaks ($I_{002}$—$I_{AM}$) and the highest intensity ($I_{002}$), while accounting for the background signal measured without the samples.

Based on the results (Figure 2), crystallinity values for untreated and mixed agro-waste were lower compared to the hydrolysed, mixed agro-waste, with both *N. mirabilis* (5000 intensity counts) and conventional (4800 intensity counts) treatments showing the highest *CrI* value. This illustrates the significant effectiveness of the *N. mirabilis* digestive fluids used to hydrolyse the lignocellulosic biomass. The agro-waste hydrolysed with *N. mirabilis* digestive fluids, show higher *CrI* values (see Table 7) which confirmed the deformation of the biomass structure when agro-waste samples were taken after 168 h, with residual highly crystalline constituents remaining in the hydrolysed agro-waste. Generally, a higher TRS extraction can be achieved, with the *CrI* value decreasing, provided further hydrolysis of

the mixed agro-waste is implemented using effective methods. In this study, the intensity counts of the water/dilute acid/cellulase (HWP/DAP/CP) hydrolysed biomass, were indicative of a higher residual crystalline phase of holocelluloses, in comparison to *N. mirabilis* hydrolysed mixed agro-waste. Two typical diffraction peaks for characterising crystallinity of ligno-holocellulosic biomass waste were observed at around 16.41 and 22.01 at 2θ, which relate to the minimum peak (101) and highest peak (002) in the lattice planes of the biomass [61]. However, the peak for the cellulose is much broader compared to other chemical compounds [42]. For biomass, the amorphous peaks normally occur at around 2θ of 18.7°, which was also observed in a study by Park et al. [61]. The crystalline values, which are shown in Table 7, were calculated based on the peak height method.

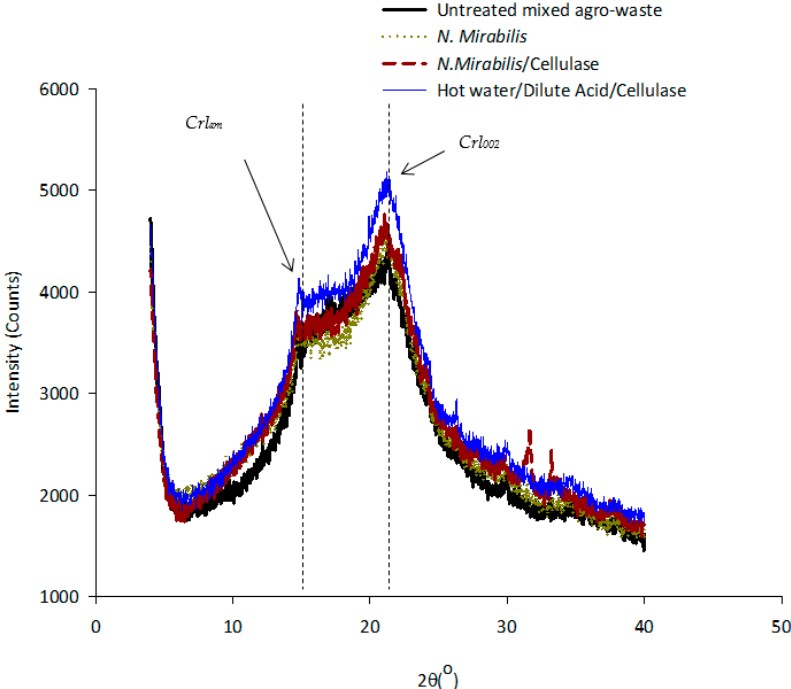

**Figure 2.** X-ray diffraction spectrum for un- and hydrolysed mixed agro-waste.

**Table 7.** Crystallinity index values of pooled hydrolysed mixed agro-waste samples after 168 h.

| Hydrolysis Methods | Crystalline Index (*Crl*) |
|---|---|
| Untreated mixed agro-waste | 15.64 |
| *N. mirabilis* [#] | 23.14 |
| *N. mirabilis*/Cellulases | 30.05 |
| Hot water/Dilute acid/Cellulases | 25.82 |

[#] Pitcher plant's digestive fluids = *Nepenthes mirabilis* digestive fluids.

### 3.3.5. FTIR Analysis for Mixed Agro-Waste

To further quantify the effectiveness of the hydrolysis method using *N. mirabilis* digestive fluids, FTIR was also used to quantify the changes in the structure and functional group distortion of the agro-waste. One of the major advantages of the FTIR is its ability to analyse samples in any state (liquid or gas) and to characterise the evolution of the substrate structures when samples are extracted from solids [62]. These advantages are also important in providing information about the substrate state (i.e., mixed agro-waste) in the FTIR spectrum [63]. The main objective of using the FTIR spectra was to assess the ability of *N. mirabilis* digestive fluids to deconstruct the biomass similarly to traditional methods, i.e., thermos chemo-biological methods, to observe the structure of mixed agro-waste constituents and functional group changes taking place in the agro-waste due to the integrated treatments process

being implemented [64]. The results obtained from the FTIR spectra (Figure 3) included a broad spectral peak at around 3324 to 3350 cm$^{-1}$ for all hydrolysed agro-waste, which is associated with the O-H stretching region reduced in the *N. mirabilis* digestive fluids' hydrolysed biomass. Furthermore, the deformations of hydrogen bonds of holocelluloses were observed at 2913 cm$^{-1}$, which illustrated the asymmetric C-H stretching regions of the mixed agro-waste for the *N. mirabilis* digestive fluids' hydrolysed agro-waste. Similar results were observed, whereby the same region, i.e., 2920 cm$^{-1}$, was indicative of the methyl and aliphatic methylene group in holocelluloses [60]. Furthermore, from 1035 to 1722 cm$^{-1}$, significant peaks associated with carbohydrates were observed, which were assigned C=C, C=O, C–H, C-O-C and C-O functional groups associated with crystalline cellulose and xylan stretching bonds, with the reduced peak prominence being attributed to the breakdown of some lattice structure of the agro-waste when *N. mirabilis* extracts are used [65]. The absorbance related with hydroxyl groups, phenolic hydroxyl group bands were observed at 1380 and 1330 cm$^{-1}$, respectively, while the absorbance related to primary hydroxyl and secondary hydroxyl groups present in lignin was observed at 1035 and 1100 cm$^{-1}$ [44]. Furthermore, this indicated the reduction of primary and secondary hydroxyl groups from the mixed agro-waste with the use of *N. mirabilis* digestive fluids for hydrolysis, showing the degradation of functional groups, thus, biodegradation of the agro-waste during hydrolysis. Additionally, other absorbance bands were observed at 1453, 2835 and 2942 cm$^{-1}$, which are related to the methoxy groups (-OCH$_3$) that are normally present in lignin [66]. In addition, absorption peaks at 1639 cm$^{-1}$ were associated with aromatic C=C vibration and C=O stretching, 1501 cm$^{-1}$ aromatic ring associated with lignin and 1410 cm$^{-1}$ attributed to C-H vibration deformation of aromatic rings in lignin, while other absorption peaks between 1100 and 1330 cm$^{-1}$ are related to ester bonds (O=C-O-C) of crystalline cellulose [67].

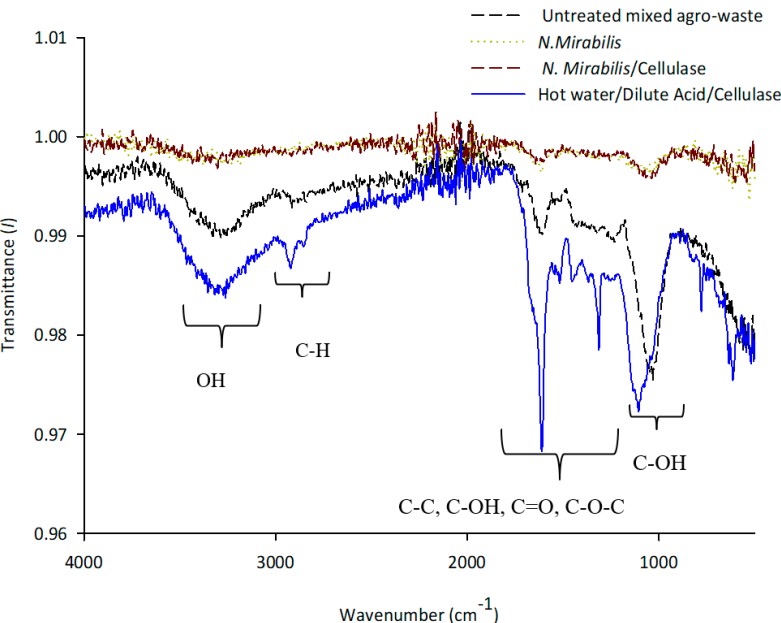

**Figure 3.** Fourier-transform infrared spectroscopy (FTIR) spectrum of pre-and un-treated mixed agro-waste.

The results obtained showed a decrease in transmittance (*I*) from $I_{2850}$, $I_{1603}$ and $I_{1035}$, with a high value of the transmittance being indicative of high crystalline structure in the residual agro-waste. Similar studies had previous confirmed that the FTIR spectral peaks can be utilised to analyse some aspects of crystallinity for different samples that contained holocellulose with the amorphous holocellulose structure being easily observable when other components of the hydrolysed agro-waste are decoupled from the lignified backbone of the waste [68]. Primarily, the hydrolysis of lignocellulosic biomass to release TRS from solid waste to a fermentable hydrolysate is large depending on functional

group decomposition. Hence, the results obtained using *N. mirabilis* digestive fluids showed lower C-C type linkages that are associated with holocelluloses. Presumptively, the holocellulose was degraded and was being released into the broth, whereby holocellulose was observed to remain on the hydrolysed solids (insoluble biomass). This has also been indicated by a higher lignin content of the residual biomass when the *N. mirabilis*/Cellulase hydrolysis process was used (see Table 3).

## 4. Conclusions

The hydrolysis of agro-waste with *N. mirabilis* 'monkey cup' digestive fluids demonstrated to be a feasible alternative hydrolysis method for TRS extraction and a phenolic compound reduction method. An alternative, single pot (batch) multi-reaction (hydrolysis) process for the pre-treatment of mixed agro-wastes, i.e., using a method that can be used to completely biovalorise a mixture of different wastes, was proposed. Additionally, by further hydrolysing/pre-treating the mixed ago-waste with cellulases, further increases in TRS and a reduction in TRPCs can be achieved, even when the process is conducted in a single pot system, using *N. mirabilis* digestive fluids as an alternative method with a suitable cocktail of enzymes that can facilitate the deligno-holocellulolytic process. The activity of carboxylesterases, xylanases and β-glucosidases activities was confirmed, which was indicative of a suitable enzymatic cocktail for hydrolysis of different waste streams, with residual biomass containing a larger proportion of lignin than holocellulose, which was extracted.

The pXRD and FTIR spectroscopy analyses demonstrated diffraction peaks related to cellulose crystallinity in the residual hydrolysed agro-waste, with a significant degradation profile from *N. mirabilis* hydrolysed agro-waste. FTIR analysis indicated the detection of the amorphous and crystalline structure of cellulose, which was similar to other studies that quantified the crystalline of cellulose. Based on the results obtained, the FTIR analysis indicated losses in functional groups. The enzymatic hydrolysis analyses also confirmed that the hydrolysis with *N. mirabilis* digestive fluids has an ability to decompose the agro-waste, with functional group prominence associated with xyan/hemicellulose and cellulose being reduced, which is indicative of agro-waste structural deformation, thus, better holocellulose extraction.

In future, it is recommended that further studies be undertaken to elucidate the different activity of the various enzymes present in the *N. mirabalis* in relation to specific biomass substrates to assess the production of individual sugar monomers, such as glucose, xylose, mannose, arabinose, galactose and rhamnose.

**Author Contributions:** Conceptualisation; N.D., J.O.A. and S.K.O.N. Draft preparations, Methodology and Investigation; N.D., E.F.I.-T. and S.K.O.N. Data interpretation, Review and Editing; B.S.C. and L.M.

**Funding:** This research was funded by the Cape Peninsula University of Technology (CPUT) and University Research Fund (URF RK 16).

**Acknowledgments:** The authors would like to acknowledge funding from the Cape Peninsula University of Technology (CPUT) and University Research Fund (URF RK 16) for financing this project. The financial assistance of the South Africa National Research Foundation (NRF) with this research is hereby acknowledged. The Pan's Carnivores Plant Nursery for allowing us to use their plants' digestive fluids. We also acknowledge the technical support of the Bioresource Engineering Research Group (*BioERG*) team and biotechnology staff.

**Conflicts of Interest:** The authors declare no conflict of interest.

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
