# Peer review of "Integrated Hydrolysis of Mixed Agro-Waste for a Second Generation Biorefinery Using Nepenthes mirabilis Pod Digestive Fluids"

_processes, doi:10.3390/pr7020064_

Reviewer 1 Report

The goal of this manuscript is initially stated as development of a single-pot lignocellulosic biomass deconstruction using N. mirabilis pod extracts. However, what the researchers actually optimized was a enzymatic pretreatment of the cellulosic components in the lignocellulosic biomass to facilitate obtaining higher amount of sugars for further conversion to chemicals or fuels. Hence, the title of the manuscript is a little misleading as the readers (just as myself) would be looking for a complete conversion of biomass to sugars in one-step. However, what they would find is an integrated N.mirabilis and cellulase hydrolysis of biomass to get sugars with the N. mirabilis not producing a high amount of total phenolic compounds that could affect the cellulase hyrolysis of the carbohydrate component in the biomass. Hence, all this study showed is a replacement for traditional thermochemical methods such as steam or hot water treatment and hence, the title and hypothesis should be modified to justify this instead. I do not think this is in any way, a single-pot conversion process, i.e., since second generation biorefinery is the application of this study. Hence, the title and the hypothesis should be appropriately modified. The language in the manuscript is very good. I did notice a few minor spelling errors such as "benignity" which is not a scientific word appropriate at its use in the manuscript.

With respect to figures and tables, they are barely adequate for justifying the process and that is actually a good part of this manuscript. However, I feel that Figure 1 is unnecessary and can be replaced with a table instead. It does not really show much in terms of valuable information or trends.

I am unsure also about the FTIR discussion since interestingly, it seems that the hot water/acid/cellulase results in lower C-C type linkages that are associated with holocellulose when compared to the N. mirabilis/cellulase process. But it looks like the total reducing sugars showed the opposite trend. I dont think it is clear and should be clearly explained as to why the current optimized "single-pot" method is better based on the FTIR data, i.e., need more targeted discussion. A high crystallinity value is indicative of cellulose but I dont know if that is really indicative of a better deconstruction process. It just allows for better access by cellulase. You can see a significant amount of research done on this by Dr. Bruce Dale at Michigan State University and Dr. Charles Wyman at University of California, Riverside. Maybe you might want to refurbish some of your discussion after referring to some of their studies (one of their study was referenced here but there are more targeted at FTIR and crystallinity during biomass deconstruction). 

Also check reference 31 for formatting issues.

Have you considered closing the mass balance on cellulose hydrolysis by also including the glucose data from the liquid hydrolysate? TRS includes other sugars but with cellulase included, the cellulose to glucose conversion is more interesting trend to look at.

I look forward to the revisions to this manuscript since they are necessary for consideration for publication in the journal. 

Author Response

Response to Reviewer 1 Comments

Point 1: I do not think this is in any way, a single-pot conversion process, i.e., since second generation biorefinery is the application of this study. Hence, the title and the hypothesis should be appropriately modified

Response 1: Thank you for this input.  We agree with the title and hypothesis adjustments. Both were modified accordingly. Please follow the track changes.

Point 2: I did notice a few minor spelling errors such as "benignity" which is not a scientific word appropriate at its use in the manuscript

Response 2:  The spelling mistakes were corrected.

Point 3: With respect to figures and tables, they are barely adequate for justifying the process and that is actually a good part of this manuscript. However, I feel that Figure 1 is unnecessary and can be replaced with a table instead. It does not really show much in terms of valuable information or trends.

Response 3: Figure 1 was replaced with a table.

Point 4: I am unsure also about the FTIR discussion since interestingly, it seems that the hot water/acid/cellulase results in lower C-C type linkages that are associated with holocellulose when compared to the N. mirabilis/cellulase process. But it looks like the total reducing sugars showed the opposite trend. I dont think it is clear and should be clearly explained as to why the current optimized "single-pot" method is better based on the FTIR data, i.e., need more targeted discussion. A high crystallinity value is indicative of cellulose but I dont know if that is really indicative of a better deconstruction process. It just allows for better access by cellulase. You can see a significant amount of research done on this by Dr. Bruce Dale at Michigan State University and Dr. Charles Wyman at University of California, Riverside. Maybe you might want to refurbish some of your discussion after referring to some of their studies (one of their study was referenced here but there are more targeted at FTIR and crystallinity during biomass

Response 4: Thank you for this input. The main objective was to use FTIR to further indicating the ability of N. mirabilis extract to deconstruct the biomass similarly to traditional methods, i.e. thermochemical methods, in order to observe the structure of mixed agro-waste constituents and chemical changes taking place in agro-waste due to integrated treatments process. We have added some more clarification related to trend of the FTIR.

Point 5: Also check reference 31 for formatting issues.

Response 5: Thank you very much for this careful analysis of the reference section. All references, including 31, were checked and corrected.

Point 6: Have you considered closing the mass balance on cellulose hydrolysis by also including the glucose data from the liquid hydrolysate? TRS includes other sugars but with cellulase included, the cellulose to glucose conversion is more interesting trend to look at.

Response 6: Thank you for this input. The mass balance on cellulose hydrolysis for glucose data was not included in our manuscript because cellulases are the cocktail of enzyme, which converts mostly celluloses and hemicelluloses to many different fermentable sugars, not just glucose. That is the reason why the study only decided to analyze total fermentable sugars (TRS) that covers all sugars relevant to bio-refinery processes. According to Sigma-Aldrich product description, this product (SAE0020) contains cellulases, ß-glucosidases, and hemicellulase, for the application of degrading cellulose to fermentable sugars. It also targets holocellulosic constituents in the agro-waste, rather than simply to produce glucose.

Reviewer 2 Report

The subject is very interesting but the quality of presentation should be improved.

The Introduction apperas confuse: a lot of concepts without a linkage.

The organization of Material and Method in subparagraph should be structured again.

For 2.2.1. A better description of plants and pod should be added.

For 2.2.2. Some lines on description of physical-chemical properties should be added and related references.

A graphial scheme of procedures described in 2.2.2.2 and  2.2.2.3

All Tables should be improved with additional detailed information.

The results should be discussed and compared in relation to previous literature data.

Also correlation of results derived by various experiments should be inserted.

The Figure of FTIR spectra should be improved.

The advantage of this technique should be better improved.

Author Response

Response to Reviewer 2 Comments

Point 1: The subject is very interesting but the quality of presentation should be improved.

Response 1: Thank for the informative comments. The overall quality has been improved. Please follow the ‘track changes’ made.

Point 1: The Introduction apperas confuse: a lot of concepts without a linkage

Response 1: The ‘quality’ and ‘flow’ of the introduction section was thoroughly improved. Please refer to the ‘track changes’ made.

Point 1: For 2.2.1. A better description of plants and pod should be added.

Response 1: A better description of the plants has been added

Point 1: For 2.2.2. Some lines on description of physical-chemical properties should be added and related references.

Response 1: The description of physical-chemical properties is now included.

Point 1: A graphical scheme of procedures described in 2.2.2.2 and  2.2.2.

Response 1: Both sections were merged and a graphical scheme of procedure was added

Point 2: All Tables should be improved with additional detailed information.

Response 2:  No specific changes were highlighted by the reviewer, therefore, minor additional information was added to the tables – as the authors do assert that details in the initial tables are adequate to describe the data, all additional information/changes are highlighted using track changes.

Point 3: The results should be discussed and compared in relation to previous literature data.

Response 3: Indeed, discussion based on previous data is key to the current study. However, very little information regarding ‘single pot multi-reaction hydrolysis process’, applications of digestive fluids, and studies on comparative analysis is available.  However, we did link our findings to some (few) partially related works. Overall, N. mirabilis (pitcher plant) digestive fluids, have not been explored for the hydrolysis of lignocellulosic biomass destined for biorefineries.

Point 4: Also correlation of results derived by various experiments should be inserted.

Response 4: We do acknowledge this important input. However, as explained above, more correlations were not done since related approaches were not attempted previously. 

Point 5: The Figure of FTIR spectra should be improved.

Response 5: The FTIR spectra has been modified and improved

Point 6: The advantage of this technique should be better improved.

Response 6: No specific technique was highlighted by the reviewer.  As many other techniques were employed in the current study, we are therefore unable to identify which technique to improve.

Point 7: Extensive editing of English language and style required

Response 6: English language and style has been edited.

Reviewer 3 Report

The subject is interesting and actual and has a direct impact in the future production of biofuels or green chemicals.

The article is well written and, in my opinion, it is almost ready to be publish.

However, minor corrections are need.

This corrections are the following:

1 - authors should try to not overkill the references used (e.g. page 2 line 26) where several references are used to sustain the same concept.

2 - The quality of figure 2 can be improved. this figure must be better explained in plain text.

3 - the same for figure 3.

Author Response

Response to Reviewer 3 Comments

Point 1: authors should try not to overkill the references used (e.g. page 2 line 26) where several references are used to sustain the same concept.

Response 1: Fewer and more relevant references are now used.

Point 2: The quality of figure 2 and 3 can be improved. this figure must be better explained in plain text.

Response 2: The quality of Figure 2 and 3 was improved and, more explanations given under the results and discussion section. Please follow the ‘track changes’ made.

Round  2

Reviewer 1 Report

The authors have made significant modifications to the manuscript and I approve the changes made. I do not agree with them on their statement about the holocellulose and the enzymatic activity on the sugars. I think TRS is, of course, a good enough to look at the overall enzymatic digestibility but it surely is better to understand the different activity of the various enzymes present in the cellulase cocktail based on this specific biomass substrate (which is not regular agricultural biomass). Glucose was only an example. Researchers usually show 5 main sugar monomers in the hyrdrolysate such as glucose, xylose, mannose, arabinose and galactose. Recently, rhamnose was added to this list. The composition of these monomers in the liquid can be compared with the composition of TRS in the solid residue (after hydrolysis) for a complete mass balance. However, the data presented in the manuscript is sufficient for publication but authors should consider the mass balance in liquid and solid, moving forward. 

Author Response

Response to Reviewer 1 Comments

Point 1: The authors have made significant modifications to the manuscript and I approve the changes made. I do not agree with them on their statement about the holocellulose and the enzymatic activity on the sugars. I think TRS is, of course, a good enough to look at the overall enzymatic digestibility but it surely is better to understand the different activity of the various enzymes present in the cellulase cocktail based on this specific biomass substrate (which is not regular agricultural biomass). Glucose was only an example. Researchers usually show 5 main sugar monomers in the hyrdrolysate such as glucose, xylose, mannose, arabinose and galactose. Recently, rhamnose was added to this list. The composition of these monomers in the liquid can be compared with the composition of TRS in the solid residue (after hydrolysis) for a complete mass balance. However, the data presented in the manuscript is sufficient for publication but authors should consider the mass balance in liquid and solid, moving forward. 

Response 1: The reviewer’s input is noted with thanks. We agree that future works will address this important mass balance issues. At the end of the conclusion the following statement was added: In future, it is recommended that further studies be undertaken to elucidate the different activity of the various enzymes present in the N. mirabalis based on specific biomass substrates to assess the production of individual sugar monomers such as glucose, xylose, mannose, arabinose, galactose and rhamnose.

Point 2: English language and style are fine/minor spell check required

Response 2:  Thorough spell checks were done (please see track changes)

Reviewer 2 Report

The advantage of FTIR technique should be marked by authors

Author Response

Response to Reviewer 2 Comments

Point 1: The advantage of FTIR technique should be marked by authors

Response 1: Thank you for this important comment. Indeed, the advantage of FTIR technique is now part of the ‘results and discussion’ section. Please refer to track changes under section 3.3.5

Point 2: English language and style are fine/minor spell check required

Response 2:  Thorough spell checks were done (please see track changes)